# Teaching, Learning and Assessing Anatomy with Artificial Intelligence: The Road to a Better Future

**DOI:** 10.3390/ijerph192114209

**Published:** 2022-10-31

**Authors:** Hussein Abdellatif, Mohamed Al Mushaiqri, Halima Albalushi, Adhari Abdullah Al-Zaabi, Sadhana Roychoudhury, Srijit Das

**Affiliations:** Department of Human & Clinical Anatomy, College of Medicine & Health Sciences, Sultan Qaboos University, Al Khoud, Muscat 123, Oman

**Keywords:** medicine, anatomy, learning, computer-aided, technology, artificial intelligence, machine learning, assessment

## Abstract

Anatomy is taught in the early years of an undergraduate medical curriculum. The subject is volatile and of voluminous content, given the complex nature of the human body. Students frequently face learning constraints in these fledgling years of medical education, often resulting in a spiraling dwindling academic performance. Hence, there have been continued efforts directed at developing new curricula and incorporating new methods of teaching, learning and assessment that are aimed at logical learning and long-term retention of anatomical knowledge, which is a mainstay of all medical practice. In recent years, artificial intelligence (AI) has gained in popularity. AI uses machine learning models to store, compute, analyze and even augment huge amounts of data to be retrieved when needed, while simultaneously the machine itself can be programmed for deep learning, improving its own efficiency through complex neural networks. There are numerous specific benefits to incorporating AI in education, which include in-depth learning, storage of large electronic data, teaching from remote locations, engagement of fewer personnel in teaching, quick feedback from responders, innovative assessment methods and user-friendly alternatives. AI has long been a part of medical diagnostics and treatment planning. Extensive literature is available on uses of AI in clinical settings, e.g., in Radiology, but to the best of our knowledge there is a paucity of published data on AI used for teaching, learning and assessment in anatomy. In the present review, we highlight recent novel and advanced AI techniques such as Artificial Neural Networks (ANN), or more complex Convoluted Neural Networks (CNN) and Bayesian U-Net, which are used for teaching anatomy. We also address the main advantages and limitations of the use of AI in medical education and lessons learnt from AI application during the COVID-19 pandemic. In the future, studies with AI in anatomy education could be advantageous for both students to develop professional expertise and for instructors to develop improved teaching methods for this vast and complex subject, especially with the increasing paucity of cadavers in many medical schools. We also suggest some novel examples of how AI could be incorporated to deliver augmented reality experiences, especially with reference to complex regions in the human body, such as neural pathways in the brain, complex developmental processes in the embryo or in complicated miniature regions such as the middle and inner ear. AI can change the face of assessment techniques and broaden their dimensions to suit individual learners.

## 1. Introduction

### 1.1. Anatomy in the Medical Profession

Anatomy has been a pillar of medical education for hundreds of years [1]. In 1235, when the first medical school in Italy was established in Salerno, anatomy was regarded as one of the most significant and significant elements of the curriculum [2]. Andrea Vesalius, the father of modern anatomy, inaugurated an era of scientific human anatomy with the publication of his principal works [2,3]. In addition, at the close of the 20th century, dissection was regarded as the foundation of medical education [2]. Anatomy is regarded as the mother of medical education due to the fact that it is the basis of all clinical medical sciences. When questioned about the significance of anatomy, doctors have ranked gross anatomy as having the greatest fundamental significance [1,4]. Numerous studies have demonstrated that medical students [1,5] and postgraduate students [1,6] consider anatomy to be of great relevance.

Medical professionals need anatomy to examine patients. Understanding sickness, diagnosing it and communicating with patients and other doctors are also crucial. Surgical and other invasive procedures require anatomical knowledge. Anatomy is crucial to diagnosis and treatment. Understanding normal anatomy and disease-induced abnormalities affects treatment [1]. Human anatomy will always be a cornerstone of medical education [7].

The teaching of anatomy has experienced numerous alterations. Time committed to anatomy instruction and a lack of personnel have long been obstacles for university pedagogical and administrative officials. Few schools adhere to traditional teaching and learning practices, whereas many have altered their curricula and adopted an integrated approach. Typically, students view anatomy lectures as dull, uninteresting and employing outdated teaching approaches. No single model or teaching method was shown to be superior to another [8] and arguably the most effective approach consisted of integrating multimodal methods into learning [9].

Figure 1 illustrates the methods of learning anatomy by any learner and these include verbal (hearing of words), auditory (responding to sounds and speech), visual (seeing objects or images), physical (going through motion), interpersonal (exchanging ideas and perspectives), logical (understanding the reasons behind) and tactile/kinesthetic (touching objects).

### 1.2. Anatomy Teaching Methods in Medical Courses

In the 20th century, with the advent of computers and other related products of the technological revolution [10], new teaching modalities have been increasingly adopted by educationalists in general and by medical educationalists in particular, which make use of computers, the internet and other multimedia devices.

Three-dimensional interactive anatomy teaching platforms were the earliest steps in utilizing computer-based technology for anatomy education. They allowed students to move between different layers of the anatomical structure and rotate the images at different angles and views. Such products include the Primal Pictures website [11] and Netters Interactive 3D anatomy [12].

More recently, a new generation of products in this category has been developed. One such product is called “digital dissection”. Here, students can perform virtual dissection by removing layers or specific structures of the body in a sequence similar to real time dissection sessions. The anatomage table is the most common example of this category of innovative anatomy teaching methods [13,14]. This technology gives the students the opportunity to virtually dissect a full-sized male and female human cadaver on a touchscreen. It was found to excite students in learning anatomy when used to supplement actual dissection. It was also suggested to be equivalent to actual dissection for some anatomical topics, such as musculoskeletal anatomy [15].

Anatomy educators took part in the utilization of the advancement in 3D printing technology. 3D printed anatomical models have become a new teaching resource in anatomy labs that can supplement other conventional resources such as dissection, prosections, and commercially available anatomical models [16]. With the enhanced availability and continuously decreasing cost of 3D printers, it is expected that this technology will be adopted more in anatomical teaching. They are useful for teaching small anatomical structures that require more detailed illustration and for illustrating anomalies and simulating surgical interventions [17,18,19].

Through proper applications, virtual reality (VR) and augmented reality (AR) technologies illustrate anatomical structures on mobile devices and headsets. While the former is completely virtual without any connection to the real world, the latter overlays the virtual structures on a real-world setting or background [20].

VR and AR present a golden opportunity for anatomists to bring anatomy teaching to a level that has never been reached before through increasing levels of interactivity and dynamic exploration of structures. They are particularly applicable for teaching structures that have complexity through permitting manipulation and multi-angle visualization of such structures [21]. Published literature shows that these technologies have been used for teaching a wide range of structures and regions of the human body [22]. However, despite the accumulating literature on the use of VR and AR in anatomy education, it seems very difficult to draw substantiated conclusions about their educational value and student engagement due to the lack of well-designed, randomized-controlled trials investigating the use of these technologies for anatomy education [23].

Another method of teaching anatomy is by body painting. Body painting is defined as “the painting of internal structures on the surface of the body with high verisimilitude” [24]. It has attracted the attention of some anatomy teachers, leading to the build-up of a small body of literature around it, though it still remains under-researched [25]. Body painting uses the powerful impact of color in aiding knowledge retention as supported by the psychology literature, through making anatomical knowledge more memorable by being an interactive process and through the high sensory stimulus it provides and the ability of color to aid recall [25]. It was also found that body painting improved students’ knowledge of anatomy. Colored body painting increases knowledge retention four weeks after the sessions in comparison to immediately after the sessions [26]. Moreover, body painting can reduce the high cost associated with more traditional cadaver-based courses [27] and other forms of living anatomy such as simulated patients [28]. It can also be an alternative for students who are overwhelmed by the emotional distress associated with dissection and who cannot cope with it [24].

Today, despite different types of curricula being adopted in various medical schools worldwide, be it traditional, problem-based learning (PBL), or an integrated type, there are some common factors with respect to the modes of teaching, learning and assessment in anatomy in all types of curricula. Though didactic lectures are still a mainstay of content delivery, other modes such as interactive tutorials, seminars, flip classrooms, team -based learning and PBL sessions with an emphasis on clinical applications have broadened the spectrum of teaching and learning in a more meaningful way [2]. Visual and computer-assisted learning (CAL) and virtual reality experiences have enhanced the relevance of anatomy to medicine, making it more evidence-based. Hence, learning has become more logical rather than mere rote learning of mere anatomical facts.

Laboratory sessions in anatomy ensure development of psychomotor domains through cadaveric dissections and microscopy skills, lay the foundation of good clinical examination skills, enhance physical endurance and prepare students for their clinical years. Students learn to approach and handle human cadavers and specimens with a humane and respectful attitude and appreciate their invaluable contribution towards anatomical science education. This develops their affective, psychological and professional domains, besides their cognitive skills [29]. Small group teaching ensures better peer interaction and peer learning, a development of team spirit and a more intimate student–teacher relationship. The pandemic compelled medical schools worldwide to innovate and adapt to online virtual classrooms or adopt blended learning. Innovation in medical and particularly the anatomical sciences, though an ongoing process, received a compelling push as medical schools worldwide had to speedily adapt to online virtual classrooms, including completely virtual dissection sessions in anatomy, or adopt blended learning during the COVID-19 crisis. Standard modes of assessment, both formative and summative, in the form of different types of theory (essays, short answer questions, multiple choice questions) and practical examinations, (objective structured practical examinations), ensured an all-round assessment to test various domains of learning. Anatomy as a basic and core medical science is a content-heavy course and every cell or tissue that exists functions and has clinical relevance. Hence, the teaching of anatomy is labor intensive. Added to that is the paucity of staff, usually overworked, thus allowing less protected time for innovations in curriculum development and research [9]. While the traditional subject-based curriculum was deemed content-heavy and not evidence-based, the integrated or PBL type of curriculum, while having certain strengths, also has disadvantages as students are prone to have certain gaps in learning with respect to anatomy, which is taught in a fragmented way across various semesters to fit into the integrated methods of learning [30]. Hence, students tend to have a ‘bird’s eye’ view and the fundamentals of basic anatomical concepts elude them, a gap that affects their clinical years. Overwhelmed with heavy course content and a disparate staff–student ratio, individual mentoring of students is becoming increasingly difficult. This has led to a decrease in the academic performance of students in many institutions. Assessments, regardless of type, may sometimes be subjective or may not be adequately designed to test all domains of learning. Currently, regardless of the pandemic, there is an ongoing need to develop new assessment strategies designed to be more comprehensive and take into account individual student diversity in learning.

## 2. Artificial Intelligence

Artificial intelligence (AI) is a scientific discipline that focuses on understanding and creating computer algorithms that are capable of performing human tasks. With the emergence of artificial neural networks and deep learning, AI has gained traction. The term itself (AI) has been well described by a few authors [31,32] and defined as a machine with intelligent behaviour and the ability to perform tasks usually performed by humans [32,33]. It is composed of three main domains: symbolic, logic-based and knowledge-based; statistical, which includes probabilistic methods and machine learning; and sub-symbolic, which includes embodied intelligence and search. All three domains can tackle different aspect of problems, including perception, planning, knowledge, reasoning and communication [33]. Humans have great hopes for AI and have started to apply it to all aspect of daily life to improve standards of living. Nowadays, AI is being used in finance, automotive engineering, economics, medicine and education [32]. 

Recently, there has been growing enthusiasm towards the application and research use of AI in medical education, as reflected by the increase in the number of publications and citations of articles related to it. AI can be implemented in all aspects of medical education, including curriculum development, curriculum analysis, learning and assessment [34]. It has the potential to identify gaps in knowledge as well as help students with specialized needs. Moreover, use of AI in medical education will improve the teaching process and enhance the students’ engagement in the learning process [35,36]. However, the implementation of AI in medical education will need medical educators to be well-equipped with a fundamental knowledge of AI. Moreover, medical curricula will require incorporation of AI into their plans to allow students to obtain the required skills, enabling them to use AI when they practice medicine. The advantages of AI in medical education are shown in Figure 2.

## 3. Lessons Learnt during the COVID-19 Pandemic

There is a continuing race between education and technology that aims to provide equitable and inclusive access to education, especially for learners with special needs and during emergencies and crisis [37]. It is very true that face-to-face communication in education is indispensable, but it cannot ensure continuation of education during circumstances that force the learners and educators to be physically distant, as in what happened during COVID-19 in 2020. Luckily, the pandemic has expedited the implementation of technology in education that had been theoretically discussed and advocated for many years [38]. This rapid implementation of digitized learning has directed more attention toward AI-driven education such as adaptive intelligent tutoring systems, virtual reality (VR), augmented reality (AR), holograms and robotics [37]. It is worth mentioning that in the pre-COVID-19 era, AI applications in education were merely used in experimental projects or to detect plagiarism [39]. 

During COVID-19 pandemic, AI-assisted learning has been proven to positively impact different aspects of the education process, including tutoring, e.g., adaptive intelligent tutoring systems, learners’ enrollment, assessment and administration tasks [39,40]. It has been shown to improve learners’ engagement, their knowledge retention and enabled them to gain and nurture new skills and aspects of human intelligence [39,41]. Currently, there is a noticeable increased interest in the adoption of AI-driven educational systems in both highly and less developed countries [39]. Scientists are predicting that, post-COVID-19, the use of digitized learning will continue and will be blended with in-person education due to the multi-level benefits it provided during the pandemics [42]. This will necessitate that future medical educators have sufficient technology background and be adaptive thinkers, in order to leverage such implementation [43]. 

## 4. Novel Uses of Artificial Intelligence to Facilitate Anatomy Teaching, Learning and Assessment

Traditionally, for centuries, since the Anatomy Act of 1832, (further modified in 1984 and as the Human Tissue Act of 2004), which legally permitted donation of a human body, either unclaimed or by the person himself or his next of kin, for the purpose of education in the health sciences, through organized dissection, the foundation of Human Anatomy as an essential stepping-stone towards providing optimum health care, was laid. Are not all medical and surgical procedures that ‘restore to normal’ a body diseased or ill at ease directed towards either restoring form (Anatomy) or its function (Physiology and Biochemistry)? Despite many computer-assisted learning aids, cadaveric dissection is still an important tool in anatomy education around the world [44]. 

The first highly labor-intensive step towards teaching and learning human anatomy as a science has been the procuring of human cadavers both for undergraduate as well as postgraduate medical education, followed by preparing these for dissection, by embalming to preserve them using hazardous chemicals, which are inhaled by the staff handling these processes over a prolonged period of time, often for decades. Additionally, the process of preparing prosected specimens both for teaching and learning anatomy and for creating interactive Anatomy museums, which have proved a valuable resource for teaching students of Anatomy, surgery and allied branches of radiology, requires expertise and adequately trained staff. To enable all this, departments of Anatomy worldwide have required abundant and expert technical staff, capable of all of the above, abundant chemical resources and equipment and cadaver storage tanks, to name a few, all of which are highly labor and cost intensive, besides requiring a large scale facility in terms of space and size.

In the last few decades, with an increasing number of medical colleges and students and a dwindling number of cadavers available for dissection, the gap between supply and demand chains has considerably widened. The occupational hazards of inhaling toxic formalin and alcohol fumes over decades, handling un-embalmed and often infected cadavers, as was highlighted in the current pandemic, by staff, is another issue of serious concern.

AI was thought to be in the realm of the future, but the future is now. It has pervaded all walks of life, from the acoustics in Alexa to accurately reporting and analyzing medical investigations, diagnostic possibilities, management outcomes and even being able to predict disease outcomes in individual patients. AI deep machine tools such as Artificial Neural Networks (ANN) or more complex Convoluted Neural Networks (CNN) are modeled on the neural networks in the human brain. Hence, they are able not only to store, process, analyze, compute and deliver data, but also to improvise their own machine learning skills over time,’ thinking’ and searching for the best options, thus functioning much like the human brain [45] and able even to distantly perform robotic surgeries. Today, AI tools with combinations of one or more of the three types of AI domains, viz. logistic, statistical and embodied intelligence, are being used in every field, including medicine. Hence, the need of the hour is to incorporate them also into the fields of health education.

It is not difficult to imagine a novel scenario where a completely automated plant or a facility affiliated to the department of Anatomy could perform all of the above mentioned tasks of procuring, preparing, embalming, dissecting, plastination and mounting anatomical specimens in a single organized facility. This would simply require a ‘non-interpretative’ type of AI facility [46]. This application of AI in the anatomical sciences, where cadaveric dissection still remains a mainstay of Anatomy and hence medical education, would greatly reduce the heavy labor-intensive load of Anatomy departments globally.

Trained expert programmers would be required to run this automated facility. An important aspect of this is that the machine would not need to be programmed to analyze or make “individual judgements” with respect to the process in general, hence a standard protocol could be established and, except for assessing any unusual situations or conditions, the manpower required would be considerably reduced. This would prove more cost-effective in the long run.

AI could be harnessed with deep learning, a sub-field of machine learning, acting as a teaching assistant, capable of analyzing body regions and segments with great accuracy and able to reproduce them for structured Anatomy demonstrations, especially for complex areas. These areas include brain, head and neck and limb musculature; e.g., the ‘Bayesian U-Net, a deep learning framework, learned the musculoskeletal Anatomy necessary to create segmentations with high accuracy in CT images [47]. This is primarily used for delineating individual muscles on CT scan images and identifying musculoskeletal disorders, but, if programmed to suit Anatomical course requirements, it could prove a valuable teaching assistant, just like the human brain, which works on the principle that, unless one knows the normal, one cannot detect the abnormal.

Interactive teaching programs such as Anatomy Chatbots [48], immediate formative assessment tools or clinical application quizzes could be modified and programmed into the deep learning AI framework to assess the extent of deeper and more logical learning and application of this knowledge in a clinical setting among students [49]. 

Anatomy as a basic medical science has always been at the core of medical education in both undergraduate and postgraduate studies. Since the first public dissection carried out by Mondino de’ Liuzzi (1275–1326) in Bologna, donor and specimen dissection have long been used as the primary methods for teaching Anatomy [50]. The best pedagogical approach to teaching Anatomy is still an unresolved issue. With the fundamental shifts in medical education’s pedagogical thinking, Anatomy, like other health science disciplines, is undergoing a major change in the curriculum with the adoption of new and novel approaches in teaching practice [9,51,52].

With the implementation of the modern integrated curricula in medical schools and with limited time devoted to Anatomy studies, the methods for teaching Anatomy need to evolve along with pedagogy change and a serious reshape [9,53,54,55,56,57,58,59,60,61].

During the COVID-19 pandemic in 2020, various governments imposed certain measures and regulations to limit its spread, causing a rapid and unexpected shift to emergency remote learning in place of face-to-face learning provision. Anatomy teaching practices have been modified accordingly and this has raised the need to seriously reshape and redesign Anatomy curricula in the future and to revise the methods for delivery of anatomical sciences [62,63,64,65].

Whatever the case may be in response to this pandemic, the methods of Anatomy education need to adapt to ensure that students achieve the required level of competency needed for their future health profession directions. The rise of AI, machine learning, virtual reality and robotics seems to be indispensable for the transformative change in anatomical education post-pandemic. These novel methods of Anatomy education are leading to profound changes in healthcare practice [66]. In addition to the proposed curricular transformation changes, the method for assessing students’ needs to be properly focused and revised to ensure that students’ knowledge and competency are appropriate [67].

The “assessment for learning” paradigm, which emphasizes active and authentic approaches for assessment, has been developed and applied broadly in medical education. In this approach of enhanced learning, learners and educators were provided with information about where they are in learning, where they need to go and how to best get there [68,69]. With recent advances in the methods for delivering Anatomy including AI, the assessment practices have adapted accordingly with modern strategies such as competency based assessment, portfolio based and programmatic assessment, which are now implemented in many modern and health-related medical curricula [51,70]. Nevertheless, the classical form of assessment with fact-based tasks is still prevalent in certain medical and health education disciplines with a wide emphasis on the summative form of assessment [71,72]. Therefore, with the advances in and reshaping of the Anatomy curricula, anatomists need to reconsider the form of assessments and apply an authentic approach that ensures appropriate application of knowledge and skills and achieves an accepted level of competency. 

Recently, with the implementation of learner-centered authentic approaches for teaching that promote active learning, assessments have improved and become future-focused as they prepare graduates to join a rapidly growing working environment and to face the continuous change caused by revolutionary technologies including machine learning, robotics and artificial intelligence [73].

Finally, with the advances in artificial intelligence and machine learning in the health field and with the use of these technologies in not only locating or collecting data but in interpreting and analyzing as well, health educators, including those in the Anatomy field, should perceive this change in educational and health-related practice and adapt it to ensure that the future practitioner is an integrator, interpreter and not only a collector of data [74,75]. Figure 3 shows details of Anatomy assessment in the era of AI. The proper application and implementation of AI in the educational process is still to be explored. 

## 5. Future Novel Scope of Artificial Intelligence in Anatomy Education

We as anatomists feel that, in the future, there is a need to there is a need to develop the following AI related novel apps or systems:(i)Virtual reality apps that will give a deeper understanding of complex branches of anatomy such as embryology and neurosciences.(ii)Analyzing, computing, storing and summarizing complex but related facts on a specific area or a subfield of Anatomy. As an example in embryology, structures that migrate and differentiate in a particular manner across human development timelines or common molecular regulation pathways which regulate their development, complex neural pathways (e.g., the direct and indirect pathways of basal ganglia) in the brain, could be visualized dynamically by a pathway ‘lighting up’ through the central nervous system, etc. The possibilities are endless. This organized information could be easily accessed through an intelligent machine tool. Thus, a complex topic could be simplified by either breaking it into segments, or making it easily accessible as an overview.(iii)AI machine tools that would ensure deep learning by enhancing visual learning with 3D images or using them in combination with 3D printing, thus creating real, accurate models of complex anatomical structures, which are sometimes difficult to dissect.(iv)Deep learning tools that would deliver structured demos of complex areas such as head, face and neck Anatomy, hence would be valuable teaching assistants.(v)AI systems that could be a valuable resource for self-review: by creating various logistic learning programs, e.g., cause–relationship types of multiple choice questions, problem based clinical scenarios for self-testing etc., a student would have access to immediate feedback and self-assessment.(vi)AI systems that could be used for objective assessment and compilation of statistical data on student scores and grades, as well as comparisons of academic performances across different cohorts and different modes of learning.(vii)For medical students and allied health science students in various phases of their courses, students could enhance their practical/clinical and diagnostic skills on interactive robots programmed to mimic specific clinical conditions in patients, which could even display responses to any ‘intervention or simulated treatment’ by medical students, thus equipping them to combat real-life crisis situations more efficiently and confidently.(viii)Using AI-based apps that could generate anatomical images and self-learning exercises in the form of more complex interactive quizzes (e.g., the Kahoot app and how it has evolved) [76]. We depicted in the schematic diagram (Figure 4) how AI can be incorporated and become beneficial for better Anatomy teaching.

According to the latest research reports, there are a few areas in Anatomy where one can find the novelty of AI and these include (1) studying human variations, (2) in healthcare practice, (3) diversity and social justice, (4) student support and (5) student learning [77]. It has also been suggested that there should be proper training and support for Anatomy educators regarding how to interpret AI student monitoring and when to depend “on the humanness of education over the data-driven recommendations” [77]. It has been suggested that focusing artificial intelligence-infused Anatomy education (AIEd) development on challenging complicated anatomical areas such as the middle ear or the pterygopalatine fossa, areas where human instructors often struggle, could be beneficial [77]. 

An ideal example of how AI in Anatomy could help in professional development was seen in the AI curriculum, which was designed around a series of lectures to provide radiology residents with a better grasp of AI algorithms [78]. In the future, all medical educational institutions set an example by which an introductory curriculum into AI in radiology, titled AI-RADS, showed a high degree of learner satisfaction in residents [78]. In fact, a variety of AI technologies are currently used in healthcare settings and they serve everything from diagnostic and detection tools to decision-support systems for patients, detecting patient anomalies, obtaining input during surgery and help in compiling the summary of clinical texts and written notes [79]. Conventional textbooks cannot always help in diagnosis and treatment. Examples are the atypical clinical features seen in dengue and COVID-19 cases, which are not even found in conventional textbooks. Clinicians are always exposed to ambiguities, anomalies and uncertainties. It is acknowledged that clinical expertise is not primarily based on objective facts and procedural decision-making but also depends on competencies even in diagnosing borderline and atypical cases [77]. 

## 6. Conclusions

To summarize, AI tools may not replace human interactions but can certainly change the face of Anatomy teaching and learning, prove a valuable resource for deep and logical learning and provide a three-dimensional virtual reality experience as close as possible to a real voyage of discovery of a living, breathing human being, thus ensuring a sound knowledge of Anatomy, the foundation of all clinical practice, medical or surgical. There are a few pertinent issues which may be raised, i.e., logistic issues such as cost of servers, training of faculty, development of tools, employment of trained technical personnel to run and maintain AI installations and apps, etc. An additional limitation may be that, even when such facilities are available, some learners may not have adequate computers or internet connectivity to access such tools. Furthermore, in order to be relevant and effective aids in Anatomy and medical education, AI-based educational programs would need to be designed not only to facilitate learning in medical and allied health science schools with varied curricula, but would also be effective in diverse cultural, racial and ethnic settings.

## Figures and Tables

**Figure 1 ijerph-19-14209-f001:**
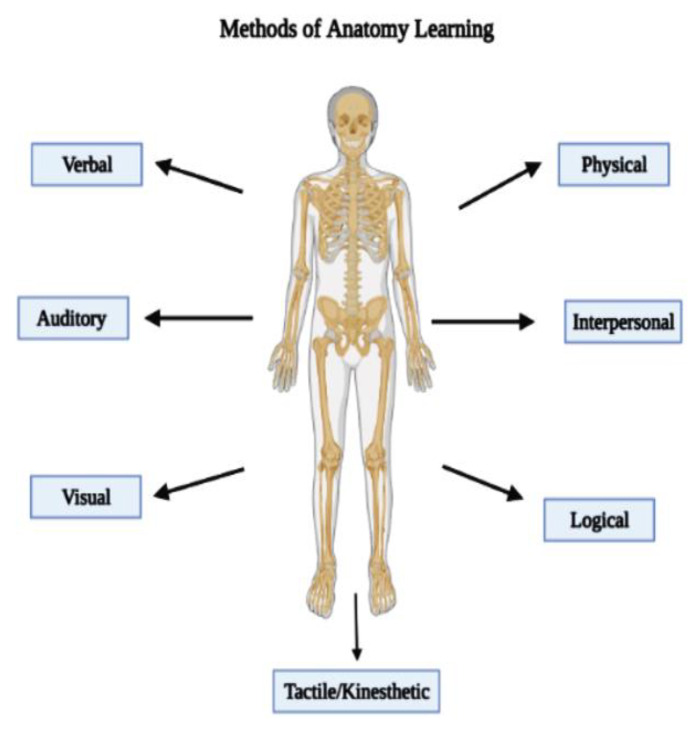
Methods of anatomy learning.

**Figure 2 ijerph-19-14209-f002:**
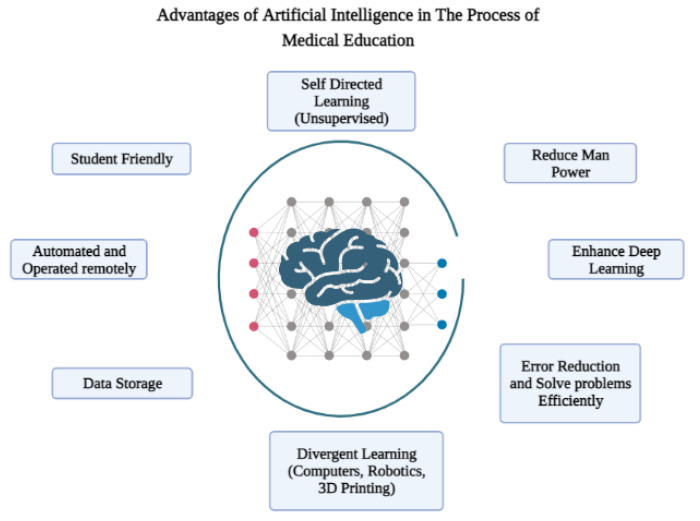
Advantages of artificial intelligence in the process of medical education.

**Figure 3 ijerph-19-14209-f003:**
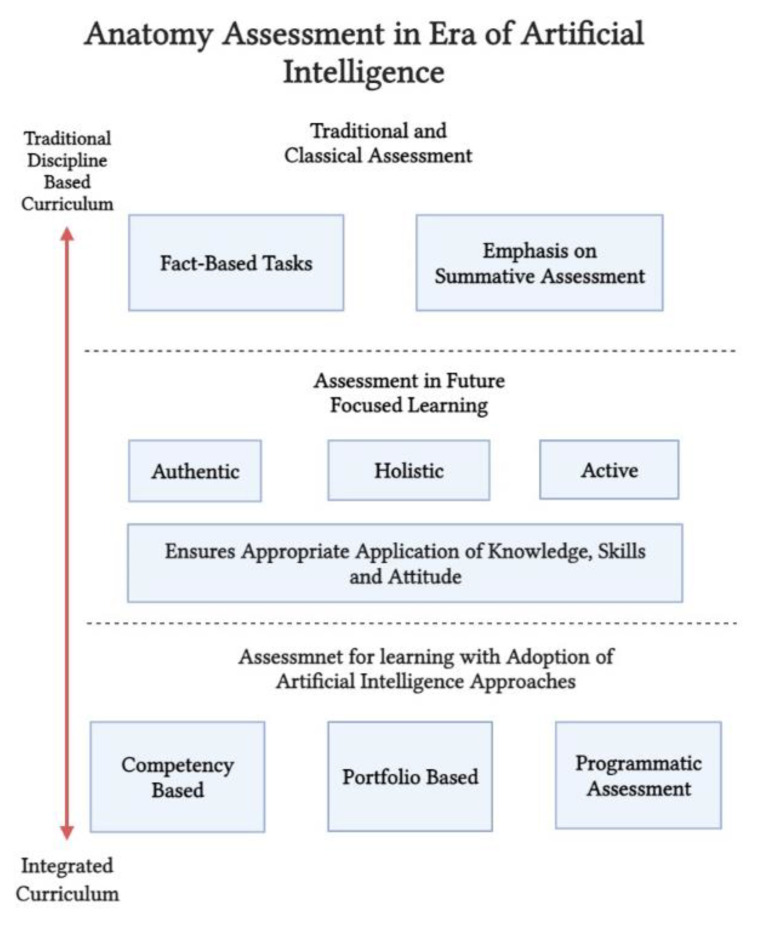
Anatomy assessment in era of artificial intelligence.

**Figure 4 ijerph-19-14209-f004:**
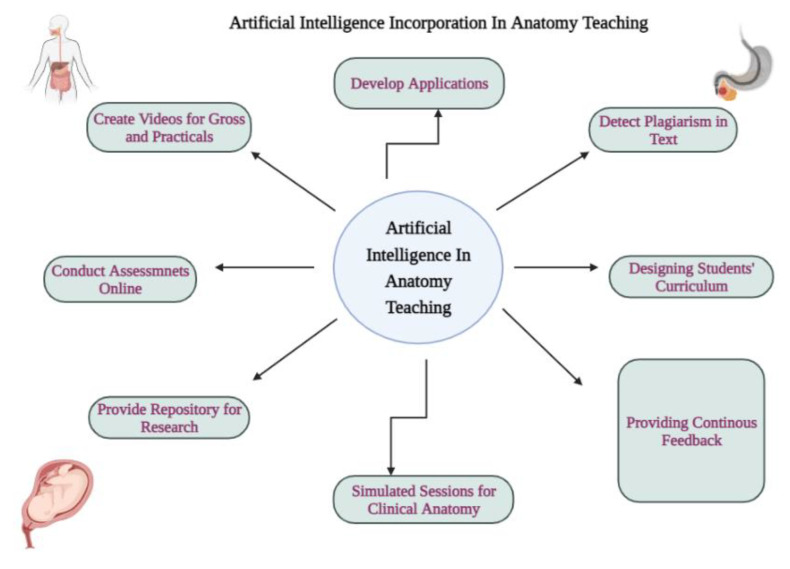
Schematic diagram showing how artificial intelligence can be incorporated in anatomy teaching.

## Data Availability

Not applicable.

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
