# Peer review of "Teaching, Learning and Assessing Anatomy with Artificial Intelligence: The Road to a Better Future"

_ijerph, 2022, doi:10.3390/ijerph192114209_

Round 1
Reviewer 1 Report
This paper presents a study on the "Role of Artificial Intelligence in Anatomy Teaching, Learning and Assessment: The Road to a Better Future". This is an interesting manuscript for this journal but I suggest a major revision. Here are some bugs in this article to help the authors to profit from this article, but if the authors can't do these comments (point by point) the article will be rejected.
================================
1) General comments:
1a) The English language is poor and should be enhanced. Please take time to improve the language. Its current version is poor.
1b) Discussion is not enough. Authors should add some technical description to the manuscript (major comment).
================================
2) Keywords:
2a) The authors must update keywords. They are old.
================================
3) Abstract:
3a) The abstract doesn’t have novelty in it. The authors should rewrite the abstract with main novelty in it.
3b) What is the main purpose of the article? The authors should focus on novelty on this section.
================================
4) Introduction and Literature Review:
4a) The introduction is very brief. The authors should extend it (some material about and novelty).
4b) I strongly recommend the authors add a new headline (1.1. literature review). At least, 6 literature review is required with more detail and their novelties (major comment).
4c) At the end of this section, the novelty of the article should be mentioned and the difference between this article and the articles they researched in this field (major comment).
================================
5) Conclusions:
5a) The authors should mention the novelty of the article and the novelty of the technique.
5b) This section should be completely rewritten and all the results of the article and also the difference between this article and other articles.
================================
6) References:
6a) References are very old, and I strongly suggested the author’s update references
Author Response
1a) The English language is poor and should be enhanced. Please take time to improve the language. Its current version is poor.
We have rectified the English language. All corrections are shown in RED color.
1b) Discussion is not enough. Authors should add some technical description to the manuscript (major comment).
It is a narrative review, so it does not have a separate discussion section. Under each section, the facts are mentioned. We have added more text on the technical aspect (shown in RED color).
================================
2) Keywords:
2a) The authors must update keywords. They are old.
The keywords were rectified as per MeSH database.
================================
3) Abstract:
3a) The abstract doesn’t have novelty in it. The authors should rewrite the abstract with main novelty in it.
The abstract has been written again. The abstract clearly highlights the latest AI incorporated for anatomy learning.
Please refer to the added sentences in the abstract- “In the present review, we highlight recent novel and advanced AI techniques such as Artificial Neural Networks (ANN), or more complex Convoluted Neural Networks (CNN), and Bayesan U-Net, which are used for teaching anatomy. We also address the main advantages and limitations of the use of AI in medical education and lessons learnt from AI application during the COVID-19 pandemic. In the future, studies with AI in anatomy education could be advantageous for both students to develop professional expertise and for instructors to develop improved teaching methods for this vast and complex subject, especially with the increasing paucity of cadavers in many medical schools. We also suggest some novel examples of how AI could be incorporated to deliver augmented reality experiences, especially with reference to complex regions in the human body, such as neural pathways in the brain, complex developmental processes in the embryo or in complicated miniature regions such as the middle and inner ear.”
3b) What is the main purpose of the article? The authors should focus on novelty on this section.
We explain the review according to the flow of facts i.e.
First of all, we start describing what the subject of anatomy is and how it is taught in the initial level after entry into medical course. Next, we describe various methods employed to teach anatomy. Then, we start with artificial intelligence (AI) and its role in medical education. Then, we highlight the lessons learnt during COVID-19 pandemic and proceed to the novelty of AI. Then, we describe the future novel scope of AI in anatomy education.
It may be mentioned that there is paucity of information on AI incorporated for anatomy teaching, learning and assessment. This is the fact which are highlighting to the readers.
We end with the conclusion where we also admit limitations. It may be mentioned that the present review is not an original paper which may have much limitations as a separate section but limitation only refers to the electronic gadgets and infrastructure needed for AI and we do mention such as constraints.
================================
4) Introduction and Literature Review:
4a) The introduction is very brief. The authors should extend it (some material about and novelty).
It may be mentioned that this is a narrative review paper. Introduction is just to introduce few facts and then we deal the subject in detail under each subheadings. All facts may not be included in the introduction but this section briefly describes what is anatomy in medical course.
We have revised all the subheadings in the manuscript.
We revised the text and all additional facts can be found under each subheading in addition to introduction section.
4b) I strongly recommend the authors add a new headline (1.1. literature review). At least, 6 literature review is required with more detail and their novelties (major comment).
All text related to literature review is included under each section. We are afraid that if we include everything under literature review section, then, it will be too crowded and the importance of each section will not be appealing to the reader.
Novelty/ novel aspects of the topic is included in relevant sections and highlighted in YELLOW color.
4c) At the end of this section, the novelty of the article should be mentioned and the difference between this article and the articles they researched in this field (major comment).
We have already stated in the abstract and in the manuscript text that there is paucity of published literature on AI and its use for teaching, learning and assessment in anatomy. Except for few papers which looked into the clinical aspect of AI, there are not enough research papers on AI and anatomy.
Only few studies incorporated for anatomy teaching in radiology and surgery disciplines. These facts warrant more studies to be conducted.
The novelty of the article can also be ascertained from the new schematic diagrams which we drew to give a better idea to any common reader. There are no past reviews with such schematic figures depicting AI and anatomy.
================================
5) Conclusions:
5a) The authors should mention the novelty of the article and the novelty of the technique.
Wherever possible in the text, we included the novelty of the study (highlighted in YELLOW).
5b) This section should be completely rewritten and all the results of the article and also the difference between this article and other articles.
Conclusion section was rewritten as per suggestion by the honorable reviewer. More facts were added (Shown in RED color).
================================
6) References:
6a) References are very old, and I strongly suggested the author’s update references
Three latest references related to AI and anatomy were added.
There are 17 references from 2021/2022 which can be found in the revised version (shown in RED color).
Reviewer 2 Report
The authors take into consideration the topic in a very correct and exhaustive way, giving a fairly broad picture of the possibilities that new technologies can offer to improve the teaching of Anatomy.
If perhaps the slightest criticism can be made of this work is the insistence that the pandemic from COVID-19 has determined a change of direction in teaching with an opening towards AI. In reality the teaching of Anatomy, for the central role that the discipline plays in Medical Education, has always sought - since the first experiences of Mondino de' Liuzzi - ways and techniques of teaching that could make up for the lack of bodies and the increasing need for knowledge.
Apart from this, the authors have been able to clearly highlight the various steps that have led technology to make the study of Anatomy increasingly comprehensible and useful. And in doing so, they recognized the potential of AI, but also the importance of face-to-face communication.
For all this and for the wealth of bibliography that denotes a wide knowledge of the subject, it is recommended the publication of the article in this form.
Author Response
The authors take into consideration the topic in a very correct and exhaustive way, giving a fairly broad picture of the possibilities that new technologies can offer to improve the teaching of Anatomy.
If perhaps the slightest criticism can be made of this work is the insistence that the pandemic from COVID-19 has determined a change of direction in teaching with an opening towards AI. In reality the teaching of Anatomy, for the central role that the discipline plays in Medical Education, has always sought - since the first experiences of Mondino de' Liuzzi - ways and techniques of teaching that could make up for the lack of bodies and the increasing need for knowledge.
Apart from this, the authors have been able to clearly highlight the various steps that have led technology to make the study of Anatomy increasingly comprehensible and useful. And in doing so, they recognized the potential of AI, but also the importance of face-to-face communication.
For all this and for the wealth of bibliography that denotes a wide knowledge of the subject, it is recommended the publication of the article in this form.
We thank the honorable reviewer for the comments. We have added more text in the revised version.
Author Response
The manuscript is a good attempt at making a case for the use of artificial intelligence (AI) in the teaching, learning and assessment of anatomy. It is quite well written, but sometimes the language is a little confusing. Some English language revision may be required. Here are some my comments on a page by page review.
We thank the reviewer for the valuable comments.
Page 2, line 57: The Figure 1 needs to be explained to some degree, either in the legend or the preceeding text.
Figure.1 is self explanatory. Still then, we have mentioned the facts in the text (Shown in YELLOW color).
Page 2, line 59: The first word should read ‘period, not ‘method’. Also, please recast the phrase in line 59 starting from ‘less time was available….’
We have deleted both the sentences as there is a separate section on COVID-19.
Page 4, line156: The new sentence should read ‘This has lead to a decrease……’
It has been rectified. Please refer to line 178, page.4.
Page 5, line 198: Change ‘barely used’ to ‘merely used’
We have rectified. Please refer to line 222, page.6.
Page 5, line 209: delete ‘to’ in ‘to have’ ad ‘to’ in ‘to be’.
Rectified. Refer to page.6, line 233.
Page 6, line 219: Change ‘may’ to ‘many’.
Rectified. Refer to line 243, page.6.
Page 7, lines 283-299 should not be under the heading ‘Benefits of AI in anatomy assessment’, because they do not talk of assessment, but AI in general. The section may start with Line 300.
We have rectified. Please to page.8, line 305 which has been merged with the later part.
Page 9, line 369-371: Isn’t this sentence meant to be (viii) under the ‘Future directions for AI’?
Yes, we agree to the suggestion and apologize for the error. We have numbered it as (viii). Please refer to line 392, page.10.
We appreciate the valuable comments by the learned reviewer.
Round 2
Reviewer 1 Report
Accept